# A Ta-TaS$_2$ monolith catalyst with robust and metallic interface for superior hydrogen evolution

Qiangmin Yu[1], Zhiyuan Zhang[1], Siyao Qiu[2], Yuting Luo[1], Zhibo Liu[3], Fengning Yang[1], Heming Liu[1], Shiyu Ge[1], Xiaolong Zou ![ORCID][1], Baofu Ding[1], Wencai Ren ![ORCID][3], Hui-Ming Cheng ![ORCID][1,3,4], Chenghua Sun ![ORCID][2,5✉] & Bilu Liu ![ORCID][1✉]

The use of highly-active and robust catalysts is crucial for producing green hydrogen by water electrolysis as we strive to achieve global carbon neutrality. Noble metals like platinum are currently used catalysts in industry for the hydrogen evolution, but suffer from scarcity, high price and unsatisfied performance and stability at large current density, restrict their large-scale implementations. Here we report the synthesis of a type of monolith catalyst consisting of a metal disulfide (e.g., tantalum sulfides) vertically bonded to a conductive substrate of the same metal tantalum by strong covalent bonds. These features give the monolith catalyst a mechanically-robust and electrically near-zero-resistance interface, leading to an excellent hydrogen evolution performance including rapid charge transfer and excellent durability, together with a low overpotential of 398 mV to achieve a current density of 2,000 mA cm$^{-2}$ as required by industry. The monolith catalyst has a negligible performance decay after 200 h operation at large current densities. In light of its robust and metallic interface and the various choices of metals giving the same structure, such monolith materials would have broad uses besides catalysis.

[1] Shenzhen Geim Graphene Center, Tsinghua-Berkeley Shenzhen Institute & Institute of Materials Research, Tsinghua Shenzhen International Graduate School, Tsinghua University, Shenzhen 518055, P. R. China. [2] College of Chemical Engineering and Energy Technology, Dongguan University of Technology, Dongguan 523808, P. R. China. [3] Shenyang National Laboratory for Materials Sciences, Institute of Metal Research, Chinese Academy of Sciences, Shenyang, Liaoning 110016, P. R. China. [4] Advanced Technology Institute, University of Surrey, Guildford, Surrey GU27XH, UK. [5] Department of Chemistry and Biotechnology, and Center for Translational Atomaterials, Swinburne University of Technology, Hawthorn, VIC 3122, Australia.
✉email: chenghuasun@swin.edu.au; bilu.liu@sz.tsinghua.edu.cn

The excessive use of fossil fuel energy has caused serious environmental problems. Hydrogen ($H_2$) is a clean energy carrier with zero-carbon emission and can be produced by water electrolysis driven by renewable energy, which is beneficial for future global carbon neutrality[1,2]. Polymer electrolyte membrane (PEM) electrolyzer technology is highly efficient and allows for high hydrogen production rates with current densities up to 2000 mA cm$^{-2}$, but suffers from problems of poor stability, high cost and low efficiency[3]. Commercial water electrolysis is usually catalyzed by noble metals like platinum (Pt) and iridium (Ir) to produce hydrogen[4], but these noble metals are scarce and have poor stability especially under large current density[5–7]. Reducing the use of noble metals or developing noble-metal-free catalysts with high activity and durability have been targeted for decades[8–13], but are far from satisfactory, especially under the large current densities demanded by industry.

Besides large current operation, in practice, stability is another key issue for hydrogen production electrodes, and is usually obtained by anchoring catalysts (such as alloying, clusters, or single-atoms[14–16]) on a conductive substrate using a binder like Nafion. With this approach, the adhesive force is usually weak and the catalysts loaded on the substrate often peel off upon hydrogen bombardment when a large operating current density is used in the hydrogen production, resulting in a short service life of the electrode[17]. Such a structure also inevitably results in a large interface resistance between the catalyst and the substrate, which slows the electron transport and causes serious Joule heating especially at large current densities[18]. As a consequence, the energy conversion efficiency is low, indicating the need to design the catalyst/substrate interface in a conceptually different way. Directly growing the catalyst on a conductive substrate could significantly improve the adhesion between them, improving the robustness of the electrode[19,20]. Such a technique, however, cannot eliminate the interface resistance, especially when catalyst-substrate interaction is dominated by van der Waals forces or ionic bonds[21,22]. Therefore, the challenge in producing such an electrode is how to achieve a high-efficiency (ultralow or even zero interface resistance) and long-durability (strong interface binding forces) hydrogen production under large current densities.

In this work, we develop a monolith catalyst (MC) to address these challenges. Specifically, a metallic transition metal dichalcogenide (m-TMDC) is vertically grown on a substrate of the same metal using an oriented-solid-phase synthesis (OSPS) method. Due to the nature of the monolith, charges can be directly transferred from the substrate to the catalyst without crossing van der Waals interfaces, providing highly efficient charge injection and an excellent HER performance. This MC has almost zero interface resistance and therefore offers unimpeded electron transfer. Moreover, the catalyst is bonded to the substrate by strong covalent bonds, which gives excellent mechanical stability to withstand the large current densities needed for efficient hydrogen production. As an example, a tantalum–tantalum sulfide (Ta-TaS$_2$) MC with a large area has been synthesized by the OSPS method and shown superior hydrogen evolution activity, achieving 2000 mA cm$^{-2}$ with a small overpotential of 398 mV and continue working for >200 h under large current densities in a 0.5 M $H_2SO_4$ electrolyte without noticeable performance decay.

## Results

### Structure and properties of the Ta-TaS$_2$ MC material. To address the problems of catalyst peel-off and large interface resistance, our strategy is to build the catalyst from the substrate as illustrated in Fig. 1a. Metallic TaS$_2$, the HER catalyst, vertically grows from Ta substrate, with strong Ta-S covalent bonding at

their interface. This structure is distinct from normal parallel stacking (see Supplementary Fig. 1), because there is no van der Waals gap at the interface between Ta and TaS$_2$. Consequently, such a structure fundamentally eliminates catalysts peel-off problem under high-current operations. More importantly, electrons do not have to tunnel over a van der Waals gap between adjacent TaS$_2$ layers to reach active sites. Overall, the design provides an ultra-strong and highly electrically conductive interface for large current density water electrolysis.

For HER, a metal substrate with abundant free electrons could inject electrons into the catalyst effectively for the subsequent reaction, and thus gives the catalyst a high reactivity[23,24]. The Gibbs adsorption free energy (G$_{H*}$) of Ta-TaS$_2$ MC was calculated. Figure 1b shows the G$_{H*}$ values of Ta, TaS$_2$, Ta-TaS$_2$ MC, and Pt, based on which Ta-TaS$_2$ MC performs similarly to Pt[25,26], with G$_{H*}$ ~0.10 eV (a G$_{H*}$ value close to zero indicates superior thermodynamic activity[27,28]), which is much better than TaS$_2$ alone (~0.61 eV). The decomposed band structures for hybrid Ta-TaS$_2$ MC were also calculated based on a model (Supplementary Fig. 1b) and are shown in Fig. 1c. We found that a large number of dispersive electronic states, contributed by Ta and TaS$_2$ jointly, cross the Fermi energy level of the system, confirming that such an interface gives excellent electrical conductivity. To understand the mechanical strength, the energy evolution $E$ with the distance $d$ between Ta and TaS$_2$ has been investigated based on parallel (Ta/TaS$_2$, Supplementary Fig. 2a) and vertical (Ta-TaS$_2$ MC, Supplementary Fig. 2b) stacking models. Using the energy $E_0$ with an equilibrium distance $d_0$ as a reference, the total energy $E$ has been calculated for a series of distances $d$ between Ta and TaS$_2$, based on which the relative energy $E-E_0$ has been derived and used as an indicator of energy cost to separate these two components from equilibrium bonded state to non-bonded state. Accordingly, larger $\Delta E = E_{d\to\infty} - E_0$ indicates stronger interaction between Ta and TaS$_2$. In our case, $\Delta E$ values of 0.37 eV and 10.56 eV are obtained for Ta/TaS$_2$ and Ta-TaS$_2$ MC, respectively, indicating more robust interface between Ta and TaS$_2$ has been built in Ta-TaS$_2$ MC than that in Ta/TaS$_2$. Accordingly, hybrid Ta-TaS$_2$ MC is expected to exhibit high reactivity, fast kinetics, and strong mechanical stability.

### Sample preparation and characterization. To test the above predictions, we synthesized a Ta-TaS$_2$ MC by an OSPS method and examined its structure. A Ta substrate with periodic holes was pre-oxidized in air (Ta→TaO$_x$), followed by oriented sulfurization along the oxidation path (TaO$_x$→TaS$_2$) and electrochemical treatment to produce a porous MC structure, as illustrated in Fig. 2a and Supplementary Information (Experimental Section). X-ray diffraction (XRD, Fig. 2b) indicates the formation of the 3R-phase TaS$_2$ with diffraction peaks of (003) at 14.9°, (101) at 32.2°, and (110) at 55.1° (PDF#89-2756)[29], which has been confirmed by the Raman spectra (Supplementary Fig. 3)[30]. X-ray photoelectron spectroscopy (XPS) measurements show four peaks for the Ta 4$f$, where the peaks at 23.3 eV and 25.2 eV are assigned to Ta$^{4+}$ in 3R-TaS$_2$ and the peaks at 26.5 eV and 27.2 eV is assigned to tantalum oxide. In addition, two S 2$p_{3/2}$ (161.9 eV) and S 2$p_{1/2}$ (162.9 eV) peaks are assigned to S$^{2-}$ in 3R-TaS$_2$ (Supplementary Fig. 4)[31]. Two factors may result in the growth of 3 R phase TMDCs. The first is the oriented-solid-phase synthesis method we used. In this method, the TaO$_x$ precursor will form first on the Ta substrate with an ordered orientation, and sulfur vapor diffuses into TaO$_x$ and converts them into sulfides. At high temperature, the chemical conversion would occur much faster than the diffusion of sulfur vapor into the TaO$_x$, thereby making sulfur diffusion the rate-limiting process.

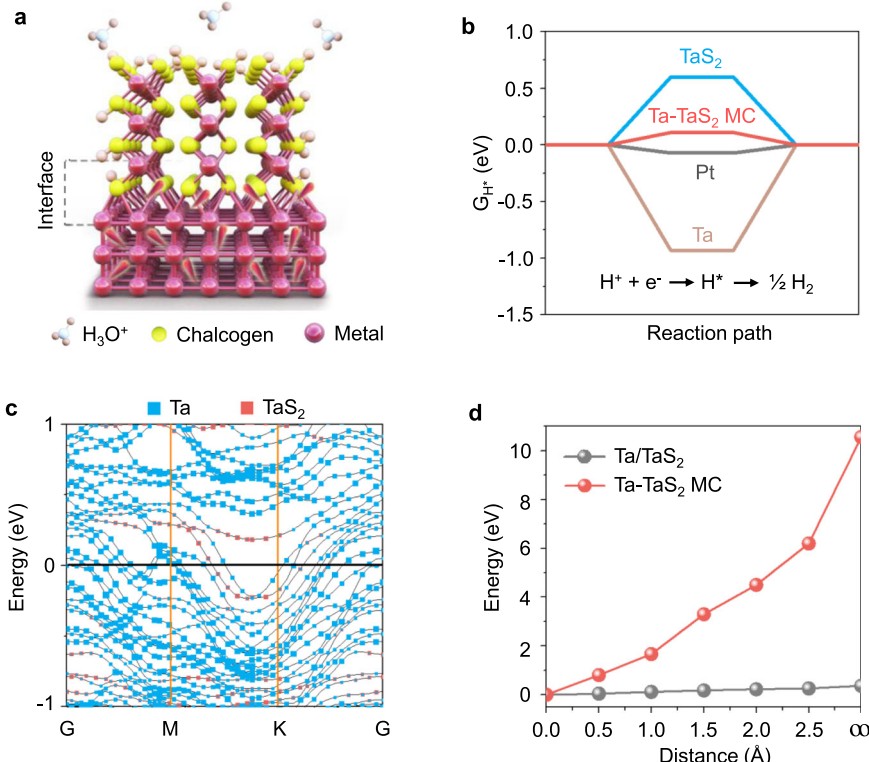

**Fig. 1 Structure and properties of the Ta-TaS₂ MC material. a** Atomic structure of the Ta-TaS₂ MC. **b** Hydrogen absorption free energy diagram of Ta, TaS₂, Ta-TaS₂ MC, and Pt catalysts. **c** Its band structures. **d** The separation energies of Ta and TaS₂ in parallel Ta/TaS₂ and vertical Ta-TaS₂ MC materials.

Diffusion along the layers through van der Waals gaps is expected to be much faster than diffusion across the layers. Such a change in growth dynamics may result in the growth of different phase materials. Second, abundant nucleation sites on Ta substrate during sulfuration process, may result in quick accumulation of layered products and the formation of 3 R phase materials.

A cross-sectional lamella of the Ta-TaS₂ MC is shown in Fig. 2c, in which TaS₂ vertically growth is seen on a Ta substrate. From high-resolution transmission electron microscopy (HRTEM) images, the TaS₂ has an interplanar distance of 0.63 nm, consistent with the (003) plane of the TaS₂ 3R-phase (see Fig. 2d and Supplementary Fig. 5)[32]. Elemental analysis of the monolith material by energy dispersive X-ray spectroscopy (EDS) elemental mapping shows that a clear interface was formed between TaS₂ and the Ta substrate (Supplementary Fig. 6). The interface was also examined by scanning TEM-high-angle annular dark field (STEM-HAADF) microscopy (Fig. 2e), with elemental Ta on both two sides while elemental S was present only on one side (Fig. 2f, g). The porosity of the MC also can be regulated by laser patterning (Supplementary Fig. 7), to provide engineerable channels for efficient mass transfer and gas diffusion[33]. From these characterizations, it is clear that a Ta-TaS₂ MC has been synthesized. The method can also be used for the synthesis of other MCs, such as niobium-niobium disulfides (Nb-NbS₂) and molybdenum-molybdenum disulfides (Mo-MoS₂). All three TMDCs have the 3R-phase structure (Supplementary Figs. 3, 4 and 8).

To investigate the mechanical properties of the interface in Ta-TaS₂ MC, a conventional Ta/TaS₂ composites (TaS₂ catalyst synthesized from Ta oxides loaded on Ta foil, where the catalyst-substrate interface has van der Waals interactions) and a Pt/C/GC composites (Pt/C catalyst pasted on a glassy carbon (GC) substrate using a Nafion binder) were used for comparison. Figure 3a shows typical force-displacement curves of the Ta-TaS₂

MC, and Ta/TaS₂ and Pt/C/GC composites. The maximum force at the top of the curve indicates the critical load of the adhesive-bonded joint before a crack starts to propagate[34]. The Ta-TaS₂ MC has an adhesive force of 39.9 N/m², which is more than three times than that of the Ta/TaS₂ composites (12.3 N/m²) and the Pt/C/GC composites (13.4 N/m²), indicating a mechanically strong interface in the MC. The electrical conductivity of the Ta-TaS₂ MC was examined to investigate its charge transfer ability in HER. As shown in Fig. 3b and Supplementary Fig. 9, an electrical conductivity of ~3 × 10⁶ S/m was obtained for the Ta-TaS₂ material which is comparable to the values for metals (Pt, Ir, Ta) and metallic TaS₂. Such a conductivity is 2–5 orders of magnitude higher than those of typical catalysts or substrates including graphite and semiconducting MoS₂, and 9 orders of magnitude higher than oxides (Supplementary Table 1), indicating excellent charge transfer kinetics across the interface between Ta and TaS₂ in the MC, as predicted by theoretical calculations. We also measured the contact angles (CAs) of the Ta-TaS₂ MC to analyze its wettability for mass transfer (Supplementary Fig. 10). The CA is 91.4° for a Ta foil and ~0° for the Ta-TaS₂ MC, indicating a good wettability of the Ta-TaS₂ MC, making it good for mass transfer in an aqueous electrolyte.

**HER performance at large current density**. Now we turn to examine the HER performance of the MC, in the hope of meeting the needs of large-area synthesis and a high-performance electrocatalyst for large current density (≥1000 mA cm⁻²) use. The Ta-TaS₂ MC has been used as a self-supporting working electrode to evaluate its catalytic performance in a 0.5 M H₂SO₄ electrolyte (Fig. 4a). It can be clearly seen from the polarization curves that for the current density to reach 2000 mA cm⁻² needs an overpotential of only 398 mV, which is much smaller than the value for Ta/TaS₂ (920 mV) and a porous Pt foil (740 mV). We used

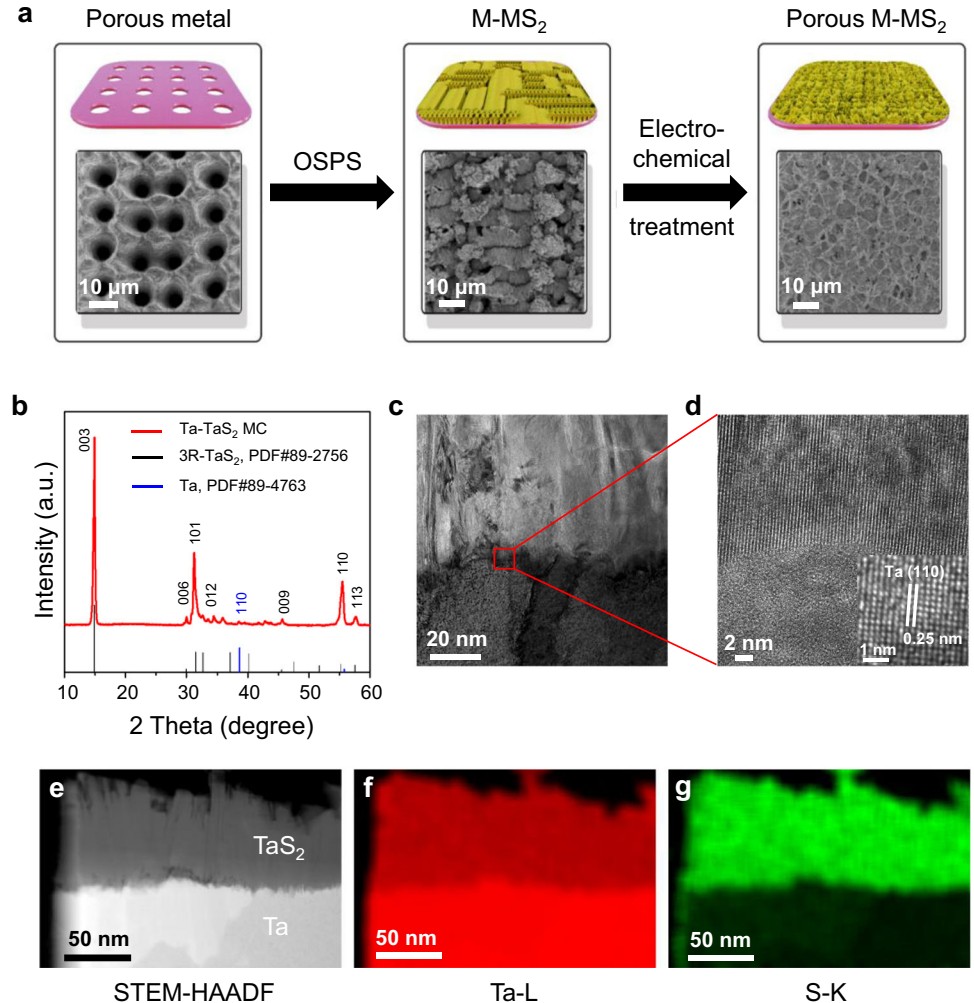

**Fig. 2 Synthesis and characterization of the Ta-TaS₂ MC. a** The OSPS synthesis process of Ta-TaS₂ MC and corresponding SEM images. **b** XRD pattern of Ta-TaS₂ MC. **c–d** TEM cross-sectional image of Ta-TaS₂ MC and a magnified image of the interface. **e** STEM-HAADF image and (**f**, **g**) corresponding STEM-EDS elemental maps based on (**f**) the Ta-L peak and (**g**) S-K peak.

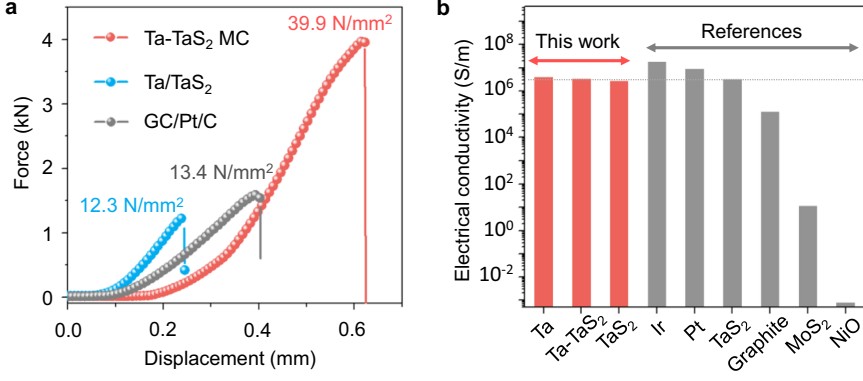

**Fig. 3 Mechanical and electrical properties of the Ta-TaS₂ MC. a** Force-displacement curves for Ta-TaS₂ MC, Ta/TaS₂, and a commercial Pt/C bound to glassy carbon for comparison. **b** Electrical conductivity of different materials.

the $\Delta\eta/\Delta\log|j|$ ratios, which has recently been proposed to evaluate the performance of an electrocatalyst over a broad range of current density[35], to evaluate the performance of catalysts at different current densities (Fig. 4b) and shows that the overpotential increases when the current increases. Both the porous Pt foil and the MC give a low ratio (~30 mV dec⁻¹) at a small current density, while their responses to current density increases are significantly

different. A sharp increase is observed for the porous Pt foil when the current density is larger than 100 mA cm⁻², and even reaches ~90 mV dec⁻¹ at $10^2$–$10^3$ mA cm⁻². For the Ta-TaS₂ MC it remains small, only ~58 mV dec⁻¹ at $10^2$–$10^3$ mA cm⁻², indicating its excellent catalytic performance at large current densities. To verify the effect of the covalently-bonded interface on catalytic performance, Ta/TaS₂ has been used as a reference and measured

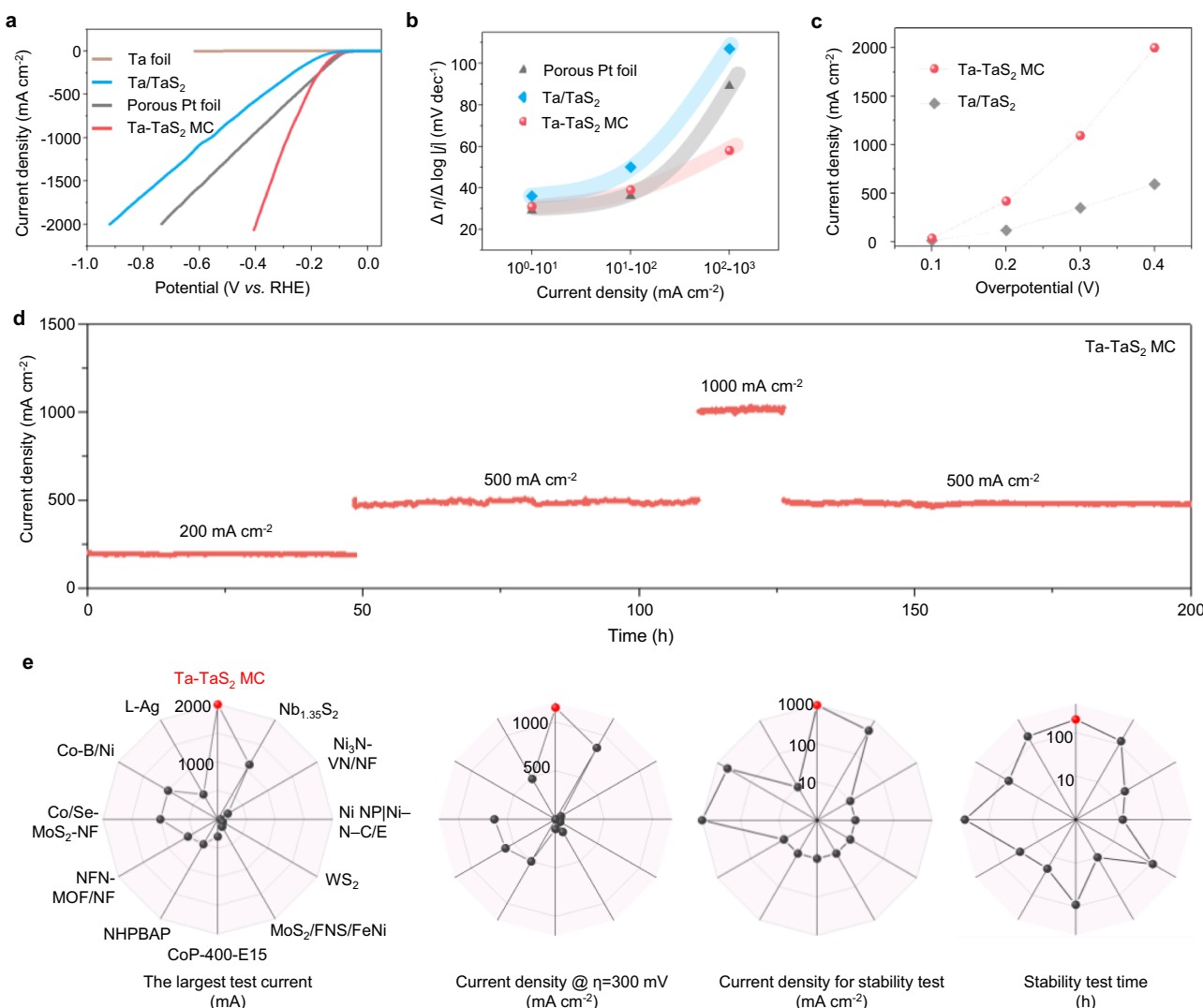

**Fig. 4 Large-current-density HER performance of the Ta-TaS$_2$ MC. a** Polarization curves of a Ta foil, Ta-TaS$_2$ MC, Ta/TaS$_2$ and a porous Pt foil measured in a 0.5 M H$_2$SO$_4$ electrolyte with a scan rate of 2 mV s$^{-1}$. **b** $\Delta\eta/\Delta\log|j|$ ratios of the Ta-TaS$_2$ MC, Ta/TaS$_2$ and porous Pt foil catalysts at different current densities. **c** HER activity of the Ta-TaS$_2$ MC and Ta/TaS$_2$. **d** i-t curves of the Ta-TaS$_2$ MC at various current densities in a 0.5 M H$_2$SO$_4$ electrolyte. **e** Comprehensive comparisons of the HER performance of Ta-TaS$_2$ MC with those reported state-of-the-art catalysts in literature. From left to right: the largest test current; the current density @$\eta$ = 300 mV; current density for stability test; stability test time.

under the same conditions, as shown in Fig. 4c. This shows that the Ta-TaS$_2$ MC always gives a much larger current density than does Ta/TaS$_2$. For example, the current density at an overpotential of 398 mV is 2000 mA cm$^{-2}$ achieved in the Ta-TaS$_2$ MC, more than three times that with Ta/TaS$_2$ (607 mA cm$^{-2}$). Similar results have also been found in the NbS$_2$ and MoS$_2$ based catalysts, i.e., the MCs show much better performance than the composite catalysts of the same metal (Supplementary Fig. 12). Given that both have TaS$_2$ as the active catalyst, the performance difference is essentially due to the interface, which is not surprising because the HER process at a large current density is overwhelmingly determined by the availability of protons and electrons. As shown above, the MC provides efficient channels for electron transfer between the catalyst and the substrate, which is essential for hydrogen production with a large current density.

The catalytic activity of the MC at a small current density was also measured to evaluate its thermodynamics. Supplementary Fig. 13 shows the polarization curves of TaS$_2$, Ta-TaS$_2$ MC and porous Pt foil catalysts, according to which the Ta-TaS$_2$ MC has a similar overpotential to Pt at a current density of 10 mA cm$^{-2}$,

indicating its high intrinsic activity. In addition, electrochemical impedance spectroscopy curves show that the Ta-TaS$_2$ MC has a charge transfer resistance of 3.2 Ω at an overpotential of 50 mV (Supplementary Fig. 14), notably lower than that of TaS$_2$ (9.0 Ω), which confirms the excellent charge transfer at the covalently-bonded interface. We therefore deduce that the interface in the Ta-TaS$_2$ MC plays a key role in HER at a large current density, which not only demonstrates high catalyst activity, but also provides an unimpeded charge transfer path between the substrate and the catalyst. Similar results have been achieved in other MCs such as Nb-NbS$_2$ and Mo-MoS$_2$ (Supplementary Figs. 13–15). We also checked the porosity of the MCs (Supplementary Figs. 16–17) and found that their activities do not have a linear relationship with the numbers of active sites, indicating the key role of the covalently-bonded interface. What more important, the Ta-TaS$_2$ MC is also durable, with no decay even at 1000 mA cm$^{-2}$ after 200 h operations, as shown in Fig. 4d and Supplementary Fig. 18. Such performance durability has been confirmed by polarization curves, XPS and XRD, neither of which show a noticeable change after 20,000 cycles (Supplementary

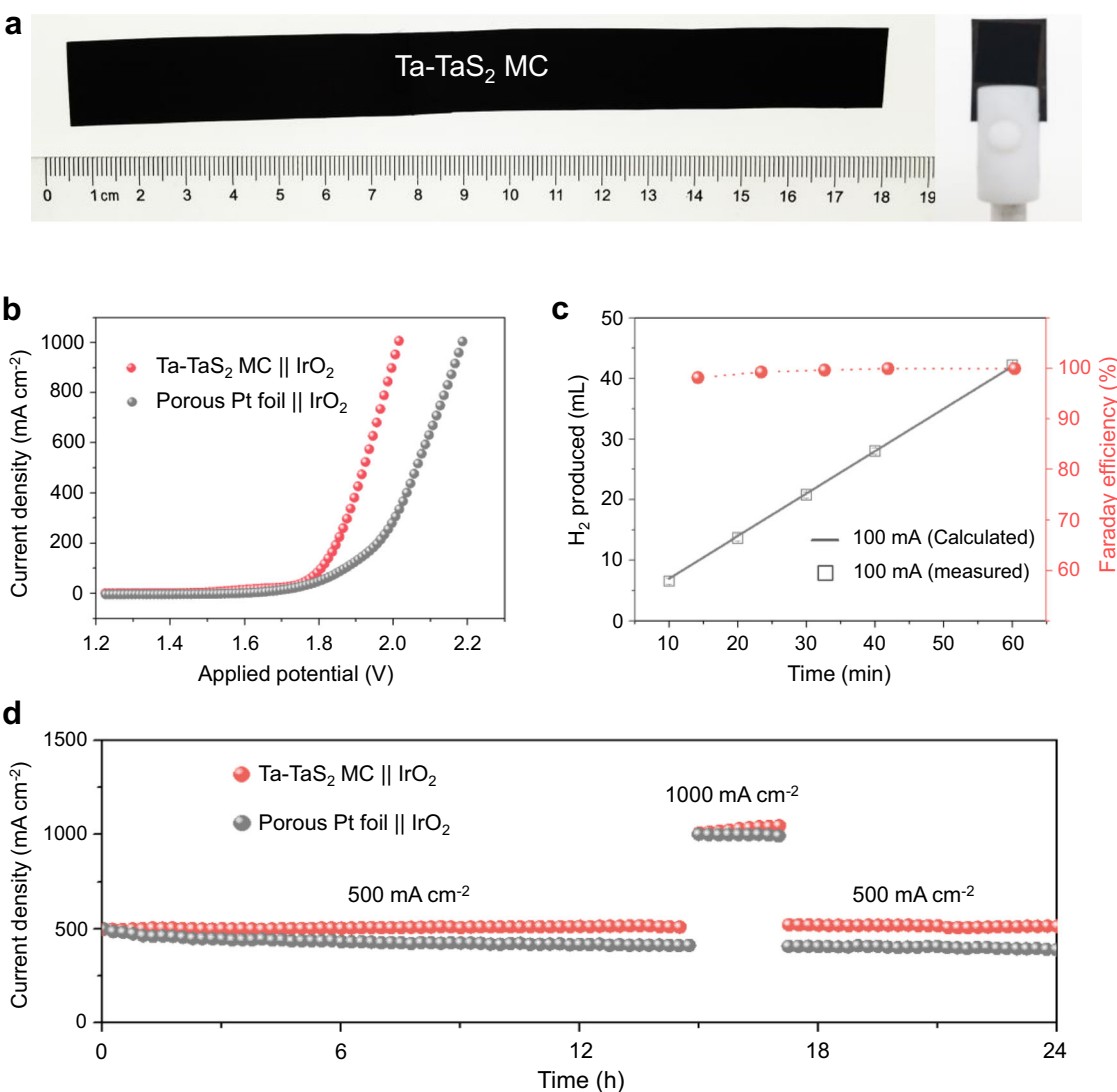

**Fig. 5 Scalable synthesis of the Ta-TaS₂ MC for PEM water electrolysis. a** A photograph of Ta-TaS₂, about 175 × 20 mm, and corresponding self-supporting electrode. **b** V-I curves of overall water electrolysis with the Ta-TaS₂ MC as the cathode and IrO₂ as the anode and the current density reaches 1000 mA cm⁻² The porous Pt foil ‖ IrO₂ were also shown for comparison. **c** Experimental and theoretical amounts of H₂ generated by the Ta-TaS₂ electrode at a fixed current density of 100 mA cm⁻² and corresponding Faraday efficiency. **d** Long-term tests of water electrolysis with Ta-TaS₂ or porous Pt foil as the cathodes, and commercial IrO₂ as the anode.

Figs. 19–21). To give a comprehensive assessment of the MC, the current density values of the Ta-TaS₂ MC at η@300 mV have been compared with other state-of-the-art HER catalysts, as shown in Supplementary Fig. 22 and Table 2. Evidently, the Ta-TaS₂ MC, with a current density of 1120 mA cm⁻², stands out from them and more importantly, the Ta-TaS₂ MC has significant advantages both from large current activity and long-term durability (Fig. 4e and Supplementary Table 3). These results show that a strong catalyst/substrate interface has been built in the MC, which can support hydrogen production at the large current density required by industry.

**Water electrolysis performance.** The production of MC can be scaled-up. As shown in Fig. 5a, a 35 cm² Ta foil, whose size is limited by the diameter of furnace, was used as a precursor to prepare the Ta-TaS₂ MC by the OSPS method. SEM images (Supplementary Fig. 23) show that the morphology of MC at different regions is similar, indicating good uniformity over a large area. We assembled the Ta-TaS₂ MC as the cathode and commercial iridium oxides (IrO₂) as the anode into a home-made electrochemical cell and studied the water electrolysis (Supplementary Fig. 24). Figure 5b shows that the reaction for the Ta-TaS₂ ‖ IrO₂ starts at around 1.50 V and reaches a current density of 1000 mA cm⁻² at 1.98 V, which is superior to that of a commercial porous Pt foil ‖ IrO₂ couple (2.20 V). H₂ and O₂ with a volume ratio close to 2:1 was collected in airtight cell (Supplementary Fig. 25), and the amount of H₂ matched well with the calculated results, indicating an almost 100% Faraday efficiency for the HER. As for its durability, it is remarkable that this electrolyzer could sustain excellent water-electrolysis with negligible decay for over 24 h when operating at large current densities of 500 and 1000 mA cm⁻². In addition to its catalytic performance, the low cost and abundance of the MC precursors are other advantages for their practical use, as metals like Ta and Nb are 2–3 orders of magnitude cheaper than Pt and their reserves are 1–3 orders of magnitude larger than Pt (Supplementary Fig. 26), making these MCs extremely promising for industrial hydrogen production by water electrolysis.

## Discussion

We have attempted to solve the challenge of large-current-density water electrolysis by the design and synthesis of MCs. The Ta-TaS$_2$ MC featured a covalently-bonded interface that not only gives it excellent mechanical strength, but also generates excellent electrical conductivity. As a result, the MC achieves an industrial current density of 2000 mA cm$^{-2}$ under a small overpotential of 398 mV. It is also durable in a strong acid electrolyte at large current densities for 200 h. For practical use, the MC coupled with commercial IrO$_2$ shows excellent performance in a water electrolyzer, with a HER current density of 1000 mA cm$^{-2}$ being achieved only by applying a potential of 1.98 V, which is superior to that of commercial Pt and IrO$_2$ couples. The MC can be prepared in large scale and at a low cost, which fills the gap between lab tests and industrial use. Because of the way the material is prepared and its high catalytic performance, the strategy described is this work may be applied to other materials or reactions to solve problems in the energy, chemistry and industrial fields.

## Methods

**Materials preparations.** The monolith catalyst (MC) was synthesized by an oriented-solid-phase synthesis (OSPS) method (pre-oxidation and oriented sul-furization of a metal substrate). The metal precursors (Mo, Nb and Ta, 99.95%) and sulfur (S, 99.0%) powders were purchased from Alfa Aesar. First, metal such as Ta foil with a size of 1 × 1 cm was treated with a laser (wavelength of 355 nm and beam size of 1–1000 μm, YT-5007, China) to produce pores with different sizes. During this process, the focused laser as a high intensity heat source (with a power of ~10$^7$ W cm$^{-2}$) was used to heat the selected area in the Ta foil, these areas were heated and vaporized, and then formed the pores. The pore sizes can be tuned by controlling the size of laser spot in the rage of 5–1000 μm. After the laser treatment, the metal foil was placed in a bath ultrasonicator to clean its surface. The cleaned porous metal precursors were transferred to a furnace and heated in air for heating 15 min to form metal oxides (MoO$_3$, Nb$_2$O$_5$ and Ta$_2$O$_5$). The heating temperatures of Mo, Nb and Ta were 300 °C, 430 °C, and 500 °C, respectively. A metal oxides and S powders were then placed in two zones of a tube furnace at temperatures of 900 °C for the Ta$_2$O$_5$ and 180 °C for S, respectively. For MoO$_3$ and Nb$_2$O$_5$, the temperature was 750 °C and 850 °C, respectively. Before sulfurization, the tube furnace was pumped down to 0.05 Torr where it was kept for 5 min before being filled to ambient pressure with Ar gas. This step was repeated twice. The reaction system was then increased to the sulfurization temperature under a mixed flow of Ar (95 sccm) and H$_2$ (5 sccm) and the growth lasted for 3 h with a mixed flow of Ar (85 sccm) and H$_2$ (15 sccm). After sulfurization, the products were cooled to room temperature slowly in an Ar gas flow (100 sccm). In this process, sulfur vapor would diffuse into TaO$_x$ which is on the top of the Ta substrate and covert TaO$_x$ into TaS$_2$. Meanwhile, the underneath Ta substrate will not react with sulfur because the protection of the TaS$_2$ layers on top. Along with the sulfurization of the top TaO$_x$ into TaS$_2$, Ta-TaS$_2$ structure made of TaS$_2$ grown on Ta would form. Finally, the porous monolith materials were prepared by electrochemical exfoliation in a three-electrode system, with a graphite rod as the counter electrode, a saturated Ag/AgCl electrode as the reference electrode, and the MC as the working electrode. Cyclic voltammetry (CV) was used to exfoliate the bulk material to produce a porous structure. The CV curve was measured between +0.10 and −0.50 V versus a reversible hydrogen evolution (RHE) with a scan rate of 50 mV s$^{-1}$ with different cycles. This process is accompanied with the gradual exposure of the active sites, resulting in a self-optimized catalytic performance. The best performance catalyst was obtained after 15,000 CV cycles.

**Materials characterizations.** The morphology and elemental analysis of the samples were investigated using a FE-SEM (HITACHI SU8010, 20 kV). TEM images were obtained by a FEI Tecnai F30 (200 kV). Raman spectra were collected on a Horiba LabRAM HR800 with a 532 nm laser excitation. XRD was carried out on a D8 Advance powder diffractometer (10°–80° 2θ, Cu Kα with a wavelength of 1.54 Å). XPS spectra were collected using a PHI 5000 Versa Probe II X-ray pho-toelectron spectrometer (Al 1486.6 eV mono at 37.0 W). The electrical resistances of the samples were measured in an ambient environment using a probe-station equipped with a semiconductor property analyzer (Model SCS 4200). The resistances of the different substrates were calculated from measuring I-V curves. The mechanical properties were measured by an electrical universal tester (MTS, C45.105).

**Electrochemical measurements.** All the electrochemical measurements were made using a three-electrode cell by a VMP300 electrochemical workstation (Biologic. Comp) in a 0.5 M H$_2$SO$_4$ electrolyte. We used a graphite rod as the counter electrode, a saturated Ag/AgCl electrode as the reference electrode, and the MC as the working electrode. We calculated the current density by electrode area (1 × 1 cm$^2$) because for industrial applications, the electrode area is of concern to evaluate the catalytic performance. Before the electrochemical tests, the H$_2$SO$_4$-electrolyte was purged with N$_2$ gas (99.999%) for 30 min. All reported potentials were calibrated by an RHE. The calibration of the saturated Ag/AgCl electrode was performed in a hydrogen saturated electrolyte of 0.5 M H$_2$SO$_4$ with the Pt foils as the working and counter electrodes, respectively. Linear sweep voltammetry (LSV) was performed at a scan rate of 0.5 mV s$^{-1}$. The average of the two potentials at which the current crossed zero was considered the thermodynamic potential for the hydrogen electrode reaction. As a result, the calibration in a 0.5 M H$_2$SO$_4$ electrolyte was based on the following equation: E$_{(RHE)}$ = E (Ag/AgCl) + 0.201 V. HER activity of the different samples was evaluated with LSV with a scan rate of 2 mV s$^{-1}$ and a small iR-compensation was applied to the data. CV was measured between +0.10 and −0.50 V versus an RHE with a scan rate of 50 mV s$^{-1}$. The stability of the catalysts was measured using a chronoamperometric method at current densities of 200, 500 and 1000 mA cm$^{-2}$. Nyquist plots were obtained at an overpotential of 50 mV while sweeping frequencies from 1 MHz to 0.1 Hz. CV measurements were performed between 0 mV and 100 mV versus a RHE at various scan rates from 10 mV s$^{-1}$ to 100 mV s$^{-1}$ to estimate the double-layer capacitance (C$_{dl}$, Supplementary Fig. 10).

**Computational methods.** Calculations were performed by spin-polarized density functional theory (DFT) using the Perdew-Burke-Ernzerhof (PBE) generalized gradient approximation (GGA) functional with the projector augmented wave (PAW) method[36–38]. Convergence accuracies of 10$^{-4}$ eV and 0.02 eV/Å for energy and force, respectively, were applied during the calculations. An energy cut-off of 450 eV was used for the expansion of the wave function. The van der Waals dispersion interaction Grimme's D3 functional was included for all models[39]. The Brillouin-zone was sampled with 3 × 3 × 1 gamma-centered k-points Monkhorst-Pack mesh for all optimizations. All the calculations were conducted with the Vienna Ab Initio Simulation Package (VASP) program[40–43]. The projector aug-mented wave (PAW) pseudopotentials were used to represent the interaction between the effective core and valence state electrons. Valence electrons of Ta (Ta 5$d^2$6$s^3$) and S (S 3$s^2$3$p^4$) were treated explicitly in the calculations. The wave-functions were expanded in plane wave basis sets with an energy cutoff of 450 eV, which was much higher than ENMAX of 223.667 and 280.000 eV for Ta and S as defined by default POTCAR provided by VASP code.

## Data availability

NA Source data are provided with this paper.

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

## Acknowledgements

We acknowledge financial support from the National Natural Science Foundation of China (Nos. 51991340, 51991343, and 51920105002), the Guangdong Innovative and Entrepreneurial Research Team Program (No. 2017ZT07C341), the Guangdong Innovation Research Team for Higher Education (2017KCXTD030), the High-level Talents Project of Dongguan University of Technology (KCYKYQD2017017), the Shenzhen Basic Research Project (Nos. JCYJ20200109144620815 and JCYJ20200109144616617), and the Economic, Trade and Information Commission of Shenzhen Municipality for the "2017 Graphene Manufacturing Innovation Center Project" (No. 201901171523).

## Author contributions

Q.Y., C.S. and B.L. conceived the idea. Q.Y. synthesized the materials, performed most of the materials characterization, and electrochemical tests. Z.Z., Y.L., F.Y., H.L. and S.G. took part in the electrochemical measurements and discussion. S.Q. and C.S. performed DFT calculations. Z.L. and W.R. performed the TEM characterization. B.L. supervised the project and directed the research. Q.Y., X.Z., B.D., H.C., C.S. and B.L. discussed and interpreted the results. Q.Y., X.Z., H.C., C.S. and B.L. wrote the paper with feedback from the other authors.

## Competing interests

The authors declare the following competing interests: Patents related to this research have been filed by Tsinghua-Berkeley Shenzhen Institute, Tsinghua University. The University's policy is to share financial rewards from the exploitation of patents with the inventors.
