## [Peer Review File · Nature Communications]

Title: A Ta-TaS₂ monolith catalyst with robust and metallic interface for superior hydrogen evolutionREVIEWER COMMENTS

Reviewer #1 (Remarks to the Author):

Electrocatalytic water splitting to produce green hydrogen is a key technology route to meet the global carbon neutralization mission. Currently developed catalysts either contain expensive Pt metals and/or exhibit unsatisfied performance, which is a major obstacle in large scale implementation of electrocatalytic hydrogen production technology. In this manuscript, the authors developed an interesting monolithic material-based catalyst with a mechanically-robust and electrically-metallic interface. Due to the metallic and covalently bonded interface, electrons can cross the interface with negligible interfacial resistance and the catalyst keep robust against hydrogen attack under large current density. Consequently, their catalyst shows superior hydrogen evolution activity and durability at impressively ultrahigh current density of 2,000 mA cm⁻², which has already met the requirement of industrial applications. The relationship between monolithic structure and catalytic performance is carefully studied based on physiochemical and mechanical properties, where the later factor is vital but has rarely been studied previously. It is thus very innovative and prospective that the authors studied this factor. Moreover, the monolith material is a family of materials, some been demonstrated in this work (like TaS₂/Ta, MoS₂/Mo, NbS₂/Nb), and more these kinds of materials could be envisioned in future. This work represents a major breakthrough and can generate big impact in the field, because it reports a type of new structure and materials, as well as their decent catalytic performance. I recommend its publication in Nature Communications after minor revisions.

Below are some technical points that the authors need to consider to further improve the paper.

1. The authors propose a mechanically robust interface, and the advantages of such kind of interface are nicely proven by theoretical and experimental results. However, the calculation methods are too brief and it would be necessary to provide the calculation parameters in detail.
2. The monolithic catalyst shows excellent stability at large current density due to the mechanical strong interface. How about the stability of the so-called Ta/TaS₂ composite catalyst? The stability of this catalyst at large current density should be studied and compared with the monolith catalyst.
3. The authors showed the performance of monolith catalyst in water electrolysis. It is known that the OER performance in water electrolysis is crucial and the resistance of solution and membrane also has a great influence on the overall performance. How about the resistance of solution and membrane in this work? Please make this point clear.
4. The authors studied three kinds of monolith materials and claimed that it is a family of materials (structures). Among these three materials, which one is the best in HER? Some brief comparisons are necessary.
5. The metallic TMDCs usually shows a self-optimized performance during HER process (see related works, e.g., Nat. Energy 2017, 2, 17127; Nat. Commun. 2017, 8, 958). Does this monolithic catalyst also show similar phenomenon? This point should be discussed.

Reviewer #2 (Remarks to the Author):

This manuscript has reported a new type of monolithic catalyst with TaS₂ vertically bonded to a conductive substrate of the same metal by strong covalent bonds. The monolithic structure provided the low resistance interface and it enhanced HER performance. However, there is insufficient evidence that TaS₂ is grown vertically on the Ta substrate. In particular, it is difficult to believe the formation of TaS₂ from the XPS data. Therefore, the current manuscript can't be accepted in Nature Communication.

1. 3R-TaS₂ is not a new material. There is a lot of literature to show the synthesis of TaS₂. (ACS Nano 2019, 13, 11874) XPS data is crucial to determine whether TaS₂ has grown or not. However, the deconvolution of XPS for 3R-TaS₂ in S4a is wrong. How can the intensity of 4f 5/2 be higher than 4f 7/2? It looks like there are two pairs of a doublet which means the material is not 3R-TaS₂. The deconvolution of S 2p in MoS₂ (Figure S4b) is also wrong. Therefore, it is hard to believe that 3R-TaS₂ is synthesized successfully because the reliability of the characterization data described by the authors is low.

2. Figure 1a makes readers confusing. There is a gap between vertically grown TaS₂ sheets, however, the STEM image of TaS₂ in figure 2 doesn't show any gap between TaS₂. Also, the STEM image for TaS₂ in figure 2 doesn't show that TaS₂ has grown vertically on Ta substrate. If the authors want to claim the formation of the vertical structure, they should provide the atomic images of the top surface of TaS₂. Any of the STEM images in this manuscript doesn't demonstrate vertical growth.

3. Figure s19 doesn't mean anything. If the authors want to show that catalyst is stable after HER measurement, they should show XPS data after 20,000 cycles.

4. The TaS₂ is very thick (over 1 micrometer in figure S6). Also, the substrate has holes. How the authors calculate the current density? The catalyst is very porous and thick. They should explain it clearly.

Reviewer #3 (Remarks to the Author):

This paper reports well-developed Ta-TaS₂ monolithic catalyst which shows excellent HER activity and superior durability under high current density (higher than 1 A/cm²) because of covalently-bonded interface between Ta and TaS₂ leading to outstanding charge transfer (electrically near-zero-resistance interface) and high mechanical stability. The Ta-TaS₂ monolithic catalysts were synthesized by OSPS (oriented-solid-phase synthesis) method. The authors demonstrated generality of this method because they synthesized Nb-NbS₂ and Mo-MoS₂.

The PEM electrolyzer technology and development of catalysts for it becomes important. In this regard, this paper has novelty and significance. This paper is technically reasonable. Thus, this reviewer recommends publication of this paper after the authors address following concerns.

1. Compare with previous reports regarding same materials, TaS₂ (Nat. Commun. 8, 958 (2017), ACS Nano 13, 11874-11881 (2019)).
2. In line 96 of page 4, they mentioned, "Fig. 1c shows the GH* values of TaS₂...". Fig. 1c should be changed to Fig. 1b. The authors should also provide Gibbs free energy for Ta for comparison.
3. In line 105 of page 4, they mentioned, "Using respect to equilibrium distance as a reference, energy cost...". The sentence is strange. Please revise it.
4. Electrically near-zero-resistance interface is a main claim of this paper. Is there any direct evidence? The data should be provided in the main text. In Fig. S13, EIS data are provided. But, there is no detailed explanation. Is it zero resistance interface? It seems not to be near-zero-resistance.
5. In page 5, they explained OSPS method including oriented sulfurization along the oxidation path (TaOx \diamond TaS₂). Based on XPS result in Fig. S4, it is not 100% sulfurization. They need to provide a portion of TaOx which will affect the HER performance. Is there any relationship between HER performance and the degree of oxidation?
6. In page 6, the vertical growth of TaS₂ on Ta is not clear to this reviewer. Is it epitaxial growth? How TaS₂ grow on Ta?
7. In Fig. 2a, porous Ta is first prepared by laser patterning. It would be good for readers that they provide how to do laser annealing in Fig. S7. In addition, the pore size affects HER performance?
8. Why 3R phase structures are synthesized in Ta-TaS₂, Nb-NbS₂, and Mo-MoS₂?
9. Table S2 and S3 are confusing. Please provide readers (or make one table) with only examples showing best performance in literature.
10. Please improve readability of Fig. 4e. It is very difficult to distinguish Ta-TaS₂ and other materials because colors are similar. Indicate specific materials instead of Reference numbers. The materials are best ones in literature?
11. Please comment on how good are HER performance of Nb-NbS₂ and Mo-MoS₂ to claim importance of the PPS method.

Response to Reviewer #1

Electrocatalytic water splitting to produce green hydrogen is a key technology route to meet the global carbon neutralization mission. Currently developed catalysts either contain expensive Pt metals and/or exhibit unsatisfied performance, which is a major obstacle in large scale implementation of electrocatalytic hydrogen production technology. In this manuscript, the authors developed an interesting monolithic material-based catalyst with a mechanically-robust and electrically-metallic interface. Due to the metallic and covalently bonded interface, electrons can cross the interface with negligible interfacial resistance and the catalyst keep robust against hydrogen attack under large current density. Consequently, their catalyst shows superior hydrogen evolution activity and durability at impressively ultrahigh current density of 2,000 mA cm⁻², which has already met the requirement of industrial applications. The relationship between monolithic structure and catalytic performance is carefully studied based on physiochemical and mechanical properties, where the later factor is vital but has rarely been studied previously. It is thus very innovative and prospective that the authors studied this factor. Moreover, the monolith material is a family of materials, some been demonstrated in this work (like TaS₂/Ta, MoS₂/Mo, NbS₂/Nb), and more these kinds of materials could be envisioned in future. This work represents a major breakthrough and can generate big impact in the field, because it reports a type of new structure and materials, as well as their decent catalytic performance. I recommend its publication in Nature Communications after minor revisions.

Response: We thank the reviewer very much for the positive recommendations. We appreciate the reviewer by writing that our work “shows superior activity”, “impressively ultrahigh current density of 2000 mA cm⁻²”, and “is innovative and prospective”.

Comment 1. The authors propose a mechanically robust interface, and the advantages of such kind of interface are nicely proven by theoretical and experimental results. However, the calculation methods are too brief and it would be necessary to provide the calculation parameters in detail.

Response 1. According to the reviewer’s suggestion, we have added detailed calculation parameters in the computational methods part in the revised SI on page 4.

“The projector augmented wave (PAW) pseudopotentials were used to represent the interaction between the effective core and valence state electrons. Valence electrons of Ta (Ta 5d²6s³) and S (S 3s²3p⁴) were treated explicitly in the calculations. The wavefunctions were expanded in plane wave basis sets with an energy cutoff of 450 eV, which was much higher than ENMAX of 223.667 and 280.000 eV for Ta and S as defined by default POTCAR provided by VASP code.”

Comment 2. The monolithic catalyst shows excellent stability at large current density due to the mechanical strong interface. How about the stability of the so-called Ta/TaS₂ composite catalyst? The stability of this catalyst at large current density should be studied and compared with the monolith catalyst.

Response 2. According to the reviewer's suggestion, we have tested the stability of the Ta/TaS₂ composite catalyst at large current density. As shown in Figure R1, the Ta/TaS₂ composite catalyst shows a significant performance decrease after 24 h test at an overpotential of 355 mV, which is much worse than Ta-TaS₂ monolith catalyst as demonstrated in Figure 4d in the manuscript.

Figure R1. I-T curves of Ta/TaS₂ composite catalyst and Ta-TaS₂ monolith catalyst. This figure was added as Figure S18 in the revised SI.

Comment 3. The authors showed the performance of monolith catalyst in water electrolysis. It is known that the OER performance in water electrolysis is crucial and the resistance of solution and membrane also has a great influence on the overall performance. How about the resistance of solution and membrane in this work? Please make this point clear.

Response 3. Yes, the OER performance is crucial in water electrolysis. The OER catalyst in our tests is IrO₂, which shows comparable performance with commercial OER catalysts. The resistance of solution and membrane also has a great influence on the overall performance as you mentioned. We have measured the resistances of solution and membrane by electrochemical impedance spectroscopy (EIS). As shown in Figure R2, the solution resistance is 0.6 Ω and the membrane resistance is < 0.5 Ω in this work, both are comparable to the reported results.

Figure R2. Electrochemical impedance measurements of the H-type cells with and w/o membrane at a constant potential of 1.55 V vs. RHE. The solution resistance is 0.6 Ω and the membrane resistance is $< 0.5 \Omega$ in this work. We have added this figure as Figure S24 in the revised SI.

Comment 4. The authors studied three kinds of monolith materials and claimed that it is a family of materials (structures). Among these three materials, which one is the best in HER? Some brief comparisons are necessary.

Response 4. This is a very good question. We have studied three kinds of monolith materials based on MoS_2 , NbS_2 , and TaS_2 . Among them, TaS_2 -based MC shows the best HER performance with an overpotential of 398 mV to reach a current density of 2000 mA cm^{-2} , which is better than that of NbS_2 - and MoS_2 - based MCs with an overpotential of 428 mV and 550 mV to reach 2000 mA cm^{-2} (Figure R3).

Figure R3. Polarization curves of Ta-TaS_2 MC, Nb-NbS_2 MC and Mo-MoS_2 MC measured in a 0.5 M H_2SO_4 electrolyte. We have added this figure as Figure S15 in the revised SI.

Comment 5. The metallic TMDCs usually shows a self-optimized performance during HER process (see related works, e.g., Nat. Energy 2017, 2, 17127; Nat. Commun. 2017, 8, 958). Does this monolithic catalyst also show similar phenomenon? This point should

be discussed.

Response 5. Yes, this monolith catalyst also shows similar self-optimized performance. In our work, the MC was synthesized by three steps, and the last step was the electrochemical treatment. In this step, the cyclic voltammetry (CV) technique was used to exfoliate the bulk TMDC materials to produce a porous structure. This process was accompanied with the gradual exposure of the active sites, resulting in a self-optimized catalytic performance (Figure R4). We used this CV technique as a synthesis step for the preparation of the best performance catalyst.

Figure R4. The self-optimized performance of a Ta-TaS₂ MC with different CV cycles. We have added some sentences in the experimental section on page 3 in the revised SI to make this point clear.

Response to Reviewer #2

This manuscript has reported a new type of monolithic catalyst with TaS₂ vertically bonded to a conductive substrate of the same metal by strong covalent bonds. The monolithic structure provided the low resistance interface and it enhanced HER performance. However, there is insufficient evidence that TaS₂ is grown vertically on the Ta substrate. In particular, it is difficult to believe the formation of TaS₂ from the XPS data. Therefore, the current manuscript can't be accepted in Nature Communication.

Response: Thank you very much for your instructive comments. We appreciate the reviewer for pointing out that our work has “reported a new type of monolith catalyst with low resistance interface and enhanced HER performance”. Regarding the evidence that TaS₂ is grown vertically on Ta and the formation TaS₂, please see below for the detailed response.

Comment 1. 3R-TaS₂ is not a new material. There is a lot of literature to show the synthesis of TaS₂. (ACS Nano 2019, 13, 11874) XPS data is crucial to determine whether TaS₂ has grown or not. However, the deconvolution of XPS for 3R-TaS₂ in S4a is wrong. How can the intensity of 4f_{5/2} be higher than 4f_{7/2}? It looks like there are two

pairs of a doublet which means the material is not 3R-TaS₂. The deconvolution of S 2p in MoS₂ (Figure S4b) is also wrong. Therefore, it is hard to believe that 3R-TaS₂ is synthesized successfully because the reliability of the characterization data described by the authors is low.

Response 1. We thank the reviewer very much for pointing this out. TaS₂ has different phases and the work the reviewer mentioned (ACS Nano 2019, 13, 11874) is 2H-phase TaS₂ (not 3R-phase), which is our previous work. In the current work, we used XRD and Raman (Figure R5) to determine the phase of TaS₂ because they are common techniques for phase analysis. Both techniques show that the TaS₂ has a typical 3R phase. Regarding XPS, we appreciate the reviewer very much for pointing this out. In the previous version, we missed Ta-O peaks when we did peak deconvolution, leading to a higher intensity of Ta 4f_{5/2} peak than Ta 4f_{7/2} peak, which is wrong as you pointed out. In addition, a pair of strong Ta-O peaks appeared in our TaS₂ sample, which might be from oxygen adsorption or oxidation during the preparation of XPS samples (Figure R6). We have taken your suggestion and carefully conducted new XPS measurements of the TaS₂, NbS₂, and MoS₂ (Figure R7). We find two oxidation peaks in TaS₂ and NbS₂, presumably due to slight oxidation because the samples were exposed in air for a few minutes (which cannot be avoided) before XPS measurements. All the peak fitting results are now reasonable with the consideration of these oxide peaks. Overall, the newly added XPS and fitted peaks suggest the formation of 3R-TMDCs in these materials, agreeing well with the XRD and Raman results. We have updated the XPS results in the revised manuscript and SI.

Figure R5. (a) XRD pattern and (b) Raman spectrum of TaS₂, showing it has a 3R-phase structure.

Figure R6. Updated Ta 4f XPS spectrum of TaS₂ in Ta-TaS₂ monolith material.

Figure R7. Updated XPS spectra of (a) 3R-TaS₂, (b) 3R-NbS₂, and (c) 3R-MoS₂.

Comment 2. Figure 1a makes readers confusing. There is a gap between vertically grown TaS₂ sheets, however, the STEM image of TaS₂ in figure 2 doesn't show any gap between TaS₂. Also, the STEM image for TaS₂ in figure 2 doesn't show that TaS₂ has grown vertically on Ta substrate. If the authors want to claim the formation of the vertical structure, they should provide the atomic images of the top surface of TaS₂. Any of the STEM images in this manuscript doesn't demonstrate vertical growth.

Response 2. We apologize for not writing this point clearly. The gap in Figure 1a represent holes (pores) in the Ta substrate. To avoid any confusion, we have followed your suggestion and removed this gap in the schematic. The updated schematic (Figure 1a) is shown below as Figure R8. Regarding the vertical growth of TaS₂ on the Ta substrate, we have provided SEM and TEM images from both top view and cross-

section view (Figure R9). The images clearly show that the TaS₂ nanosheets with a (003) layer spacing of 0.63 nm vertically standing on Ta substrate.

Figure R8. Updated atomic structure of the Ta-TaS₂ monolith catalyst. This scheme was used as updated Figure 1a.

Figure R9. The SEM (a-b) and TEM (c-d) images of Ta-TaS₂ monolith material from top view and cross-sectional view. From the cross-sectional TEM view in d, we can clearly see that TaS₂ is grown vertically on the Ta substrate.

Comment 3. Figure s19 doesn't mean anything. If the authors want to show that catalyst is stable after HER measurement, they should show XPS data after 20,000 cycles.

Response 3. We have followed the reviewer’s nice suggestion and conducted XPS measurements of TaS₂ after 20,000 cycles. As shown in Figure R10, the two samples show similar XPS peaks, revealing its stability.

Figure R10. The Ta 4f XPS spectra of Ta-TaS₂ MC before and after 20,000 cycles. We have added this Figure as Figure S20b in the revised SI.

Comment 4. The TaS₂ is very thick (over 1 micrometer in figure S6). Also, the substrate has holes. How the authors calculate the current density? The catalyst is very porous and thick. They should explain it clearly.

Response 4. This is a very good point. We calculated the current density by electrode area (1×1 cm²) in this manuscript. This work aims for the industrial application under large current density, in which the electrode area is of concern and is usually used to evaluate the catalytic performance. We have added a sentence in electrochemical measurements part on page 3 in the revised SI to make this point clear. “We calculated the current density by electrode area (1×1 cm²) because for industrial HER applications, the electrode area is of concern to evaluate the catalytic performance”.

Response to Reviewer #3

This paper reports well-developed Ta-TaS₂ monolithic catalyst which shows excellent HER activity and superior durability under high current density (higher than 1 A/cm²) because of covalently-bonded interface between Ta and TaS₂ leading to outstanding charge transfer (electrically near-zero-resistance interface) and high mechanical stability. The Ta-TaS₂ monolithic catalysts were synthesized by OSPS (oriented-solid-phase synthesis) method. The authors demonstrated generality of this method because they synthesized Nb-NbS₂ and Mo-MoS₂. The PEM electrolyzer technology and development of catalysts for it becomes important. In this regard, this paper has novelty and significance. This paper is technically reasonable. Thus, this reviewer recommends publication of this paper after the authors address following concerns.

Response. We thank the reviewer very much for the positive recommendations. We also appreciate the reviewer by writing that our work “shows excellent activity and

superior durability under high current density”, “outstanding charge transfer”, “high mechanical stability”, as well as the work “has novelty and significance”.

Comment 1. Compare with previous reports regarding same materials, TaS₂ (Nat. Commun. 8, 958 (2017), ACS Nano 13, 11874-11881 (2019)).

Response 1. According to the reviewer’s suggestion, we have compared catalytic performance with previous reports regarding the same TaS₂ materials. As shown in Table R1, our Ta-TaS₂ material shows the best performance in terms of overpotential (η) at 10 mA cm⁻², Tafel slopes, and stability.

Table R1. A comparison of the HER performance of TaS₂ materials. This table was added as Table S2 in the revised SI.

Samples (Phase)	η @10 mA cm ⁻² (mV)	Tafel slopes (mV dec ⁻¹)	Stability	References
TaS ₂ (2H)	65-150	45-49	N/A	Nat. Commun. 8, 958 (2017)
Au/TaS ₂ (2H)	101	53	12 h @ 10 mA cm ⁻²	ACS Nano 13, 11874-11881 (2019)
Ta-TaS ₂ (3R)	65	30	200 h @ 200-1000 mA cm ⁻²	This work

Comment 2. In line 96 of page 4, they mentioned, “Fig. 1c shows the G_{H*} values of TaS₂...”. Fig. 1c should be changed to Fig. 1b. The authors should also provide Gibbs free energy for Ta for comparison.

Response 2. We thank the reviewer for pointing out this typo. We have corrected it and updated this sentence in the revised manuscript. According to the reviewer’s suggestion, we have also calculated the Gibbs free energy of Ta for comparison (Figure R11). The Gibbs free energy of Ta is -0.93 eV, which is worse than that of Ta-TaS₂ MC and Pt catalysts.

Figure R11. Hydrogen absorption free energy diagram of Ta, TaS₂, Ta-TaS₂ MC and Pt catalysts. This figure was added as Figure 1b in the revised manuscript.

Comment 3. In line 105 of page 4, they mentioned, “Using respect to equilibrium

distance as a reference, energy cost...”. The sentence is strange. Please revise it.

Response 3. Thank you for your careful reading, and we have revised this sentence as follows.

“Using the energy E_0 with an equilibrium distance d_0 as a reference, the total energy E has been calculated for a series of distances d between Ta and TaS₂, based on which the relative energy $E-E_0$ has been derived and used as an indicator of energy cost to separate these two components from equilibrium bonded state to non-bonded state. Accordingly, larger $\Delta E=E_{d \rightarrow \infty}-E_0$ indicates stronger interaction between Ta and TaS₂. In our case, ΔE values of 0.37 eV and 10.56 eV are obtained for Ta/TaS₂ and Ta-TaS₂ MC, respectively, indicating more robust interface between Ta and TaS₂ has been built in Ta-TaS₂ MC than that in Ta/TaS₂.”

Comment 4. Electrically near-zero-resistance interface is a main claim of this paper. Is there any direct evidence? The data should be provided in the main text. In Fig. S13, EIS data are provided. But, there is no detailed explanation. Is it zero resistance interface? It seems not to be near-zero-resistance.

Response 4. This is very important point. You are correct that it is a near-zero resistance interface, not near-zero-resistance. The electrically near-zero-resistance interface we claimed is the interface between Ta substrate and TaS₂ materials. The evidence to support this claim was provided in Figure 3b in the manuscript, where we measured the conductivity of Ta, TaS₂, and Ta-TaS₂ monolith material. A conductivity of $\sim 3 \times 10^6$ S/m was obtained for the Ta-TaS₂ material, which is comparable to the values for Ta metal (4×10^6 S/m) and metallic TaS₂ (2.8×10^6 S/m) material. Therefore, it is a near-zero resistance interface between Ta substrate and TaS₂ in Ta-TaS₂ monolith catalyst. Regarding the EIS data in Fig. S13, it is used to evaluate the catalytic performance of the Ta-TaS₂ monolith catalyst during HER process. The results showed a solution resistance of $\sim 0.2 \Omega$ and a charge transfer resistance of $\sim 3.2 \Omega$, which confirms the excellent charge transfer for Ta-TaS₂ monolith catalyst. Based on the reviewer’s suggestion, we have provided the above detailed explanation about this data on pages 9 and 12 in the revised SI.

Comment 5. In page 5, they explained OSPS method including oriented sulfurization along the oxidation path (TaO_x→TaS₂). Based on XPS result in Fig. S4, it is not 100% sulfurization. They need to provide a portion of TaO_x which will affect the HER performance. Is there any relationship between HER performance and the degree of oxidation?

Response 5. Thank you for this inspiring suggestion. We have taken this suggestion and compared samples with different portions of TaO_x (including fully-sulfurized samples and partially-sulfurized samples). Because XPS is a surface sensitive technique, which might not reflect the TaO_x portion accurately, here we used XRD to characterize the samples. As shown in Figure R12, for the fully-sulfurized sample, no TaO_x can be seen in the XRD pattern (Figure R12a), while several TaO_x diffraction peaks (e.g., 22.9°, 28.3°) can be seen in the partially-sulfurized samples (Figure R12b). Our

electrochemical measurements show that the fully-sulfurized samples show much better HER performance than the partially-sulfurized samples (Figure R13). Therefore, we chose the former in this research.

Figure R12. The XRD patterns of (a) Ta-TaS₂ monolith materials with fully-sulfurized and (b) partially-sulfurized structures.

Figure R13. Polarization curves of Ta-TaS₂ monolith materials with fully-sulfurized (red line) and partially-sulfurized (black line) structures.

Comment 6. In page 6, the vertical growth of TaS₂ on Ta is not clear to this reviewer. Is it epitaxial growth? How TaS₂ grow on Ta?

Response 6. We have followed your suggestion and conducted additional SEM and TEM characterization of the sample from the top-view and cross-section view. The images clearly show that the TaS₂ nanosheets with a (003) layer spacing of 0.63 nm vertically standing on Ta substrate (Figure R14b, d). Currently we do not know whether it is epitaxial growth, which needs more characterization at atomic resolution. Regarding how TaS₂ grown on Ta, we propose the following process. The TaS₂ was grown on Ta by an OSPS method. In this process, sulfur vapor would diffuse into TaO_x which is on the top of the Ta substrate and convert TaO_x into TaS₂. Meanwhile, the underneath Ta substrate will not react with sulfur because the protection of the TaS₂ layers on top. Along with the sulfurization of the top TaO_x into TaS₂, Ta-TaS₂ structure made of TaS₂ grown on Ta would form. We have added the above growth mechanism

on page 2 in the revised SI.

Figure R14. The SEM (a-b) and TEM (c-d) images of Ta-TaS₂ monolith material from top view and cross-sectional view. From the cross-sectional TEM view in d, we can clearly see that TaS₂ is grown vertically on the Ta substrate.

Comment 7. In Fig. 2a, porous Ta is first prepared by laser patterning. It would be good for readers that they provide how to do laser annealing in Fig. S7. In addition, the pore size affects HER performance?

Response 7. We have followed your suggestion and provided detailed information for laser patterning technique in revised SI on page 2 (Experimental Section) The detailed information is as follows.

“First, the metal such as Ta foil with a size of 1 cm×1 cm was treated with a laser (wavelength of 355 nm and beam size of 1-1000 μm, YT-5007, China) to produce pores with different sizes. During this process, the focused laser as a high intensity heat source (with a power of $\sim 10^8$ W cm⁻²) was used to heat the selected area in the Ta foil, these areas were heated and vaporized, and then formed the pores. The pore sizes can be tuned by controlling the size of laser spot in the range of 5 to 1000 μm. After the laser treatment, the metal foil was placed in a bath ultrasonicator to clean its surface.”

Regarding the effect of pore size on HER performance, we found that samples with 10 μm and 20 μm pores have slightly better performance than those with 50 μm pores. In addition, samples with pores have much better performance than those without pores (Figure R15). In this work, the pore size is 10 μm.

Figure R15. Polarization curves of Ta-TaS₂ MC samples with different pore sizes. This is the Figure S16a in the SI.

Comment 8. Why 3R phase structures are synthesized in Ta-TaS₂, Nb-NbS₂, and Mo-MoS₂?

Response 8. It is indeed very interesting that our materials show 3R phase, which is different with most reports where 2H phase materials are grown. We have summarized two factors. One factor is the unique oriented-solid-phase synthesis method we used. In this method, the TaO_x precursor was first bonded on the Ta substrate with an ordered orientation, and sulfur vapor diffuses into TaO_x and converts them into sulfides. At high temperature, the chemical conversion occurs much faster than the diffusion of sulfur gas into the TaO_x, thereby making sulfur vapor diffusion the rate-limiting process. Diffusion along the layers through van der Waals gaps is expected to be much faster than diffusion across the layers. Such a change in growth dynamics may result in the growth of different phase materials. Another factor is the abundant nucleation sites on Ta substrate during sulfuration process, may result in quick accumulation of layered products and the formation of 3R phase materials. Similar mechanism has also been reported in literature (Nano Letter 2013, 13, 1341-1347; PNAS 2013, 110, 19701-19706). We have added the above discussions on page 6 in the revised manuscript.

Comment 9. Table S2 and S3 are confusing. Please provide readers (or make one table) with only examples showing best performance in literature.

Response 9. According to the reviewer's suggestion, we have summarized these data in one table (Table R2), which is now added as Table S3 in the revised SI.

Table R2. Comprehensive comparisons of the HER performance of Ta-TaS₂ MC obtained in this work with those reported in literature.

Catalysts	The largest test current (mA)	Current density(mA cm ⁻²) ($\eta=300$ mV)	The stability test current density (mA cm ⁻²)	The stability test time (h)	Refs
Co/Se-MoS ₂ -NF	1000	~ 630	1000	360	Ref. S16
NFN-MOF/NF	600	~ 595	100	30	Ref. S19
Ni ₃ N-VN/NF	200	~ 120	10	20	Ref. S21
CoP-400-E15	500	~ 100	500	90	Ref. S22
MoS ₂ /FNS/FeNi	150	~ 150	10	10	Ref. S24
NHPBAP	500	~ 500	100	20	Ref. S25
Strained WS ₂	100	~ 55	30	120	Ref. S26
L-Ag	500	~ 480	10	160	Ref. S27
2H-Nb _{1.35} S ₂	1000	~ 800	500	130	Ref. S30
Ni NP Ni-N-C/E	50	~32	30	10	Ref. S31
Co-B/Ni	500	~165	50	20	Ref. S17
Ni-W	700	~ 500	10	30	Ref. S18
NCN-1000-5	50	~ 50	20	~4	Ref. S20
2H-TaS ₂	160	~100	N/A	N/A	Ref. S28
2H-NbS ₂	200	N/A	N/A	N/A	Ref. S11
1T-TaS ₂	120	~ 22	30	24	Ref. S29
Nb-NbS ₂ MC	2000	995	N/A	N/A	This work
Ta-TaS ₂ MC	2000	1115	200-1000	200	This work

Comment 10. Please improve readability of Fig. 4e. It is very difficult to distinguish Ta-TaS₂ and other materials because colors are similar. Indicate specific materials

instead of Reference numbers. The materials are best ones in literature?

Response 10. According to the reviewer's suggestion, we have divided this Fig. 4e into four figures as follows (Figure R16), which it is easy to distinguish. In addition, we have added the specific materials in each figure as you suggested. To our best knowledge, these specific materials are the best ones in literature reported so far.

Figure R16. Comprehensive comparisons of the HER performance of Ta-TaS₂ MC with those reported state-of-the-art catalysts in literature. From left to right: the largest test current; the current density @ $\eta=300$ mV; current density for stability test; stability test time. We have added this figure as Figure 4e in the revised manuscript.

Comment 11. Please comment on how good are HER performance of Nb-NbS₂ and Mo-MoS₂ to claim importance of the OSPS method.

Response 11. To highlight the importance of the OSPS method, two additional samples have been synthesized and compared. One is the monolith catalyst such as Nb-NbS₂ MC and Mo-MoS₂ MC synthesized by the OSPS method, another is the composite catalyst such as Nb/NbS₂ and Mo/MoS₂ synthesized by simple sulfuration. The electrochemical results in Figure R17 showed that the Nb-NbS₂ MC and Mo-MoS₂ MC achieved a current density of 1750 mA cm⁻² and 826 mA cm⁻² at the overpotential of 400 mV, which is much larger than that of the Nb/NbS₂ (463 mA cm⁻²) and Mo/MoS₂ (242 mA cm⁻²) at the same overpotential. The excellent performance of these MCs synthesized by the OSPS method is stems from the excellent charge transfer kinetic in MC and robust interface between substrate and catalysts. These results support the importance of the OSPS method in synthesizing high performance Nb-NbS₂ and Mo-MoS₂ based monolith catalysts.

Figure R17. Polarization curves of different samples including Nb-NbS₂ MC, Nb/NbS₂ composite, Mo-MoS₂ MC, and Mo/MoS₂ composite measured in a 0.5 M H₂SO₄ electrolyte. We have added this figure as Figure S12 in the revised SI.

REVIEWERS' COMMENTS

Reviewer #1 (Remarks to the Author):

I recommend its publication in Nature Communications in this revisions.

Reviewer #2 (Remarks to the Author):

The authors have addressed well about reviewers' comments. This is revised manuscript can be accepted.

Reviewer #3 (Remarks to the Author):

The authors addressed all concerns raised by this reviewer. Thus, this paper is recommended for publication.

Response to Reviewers' Comments

Reviewer #1:

I recommend its publication in Nature Communications in this revisions.

Response: We thank the reviewer very much for your recommendation.

Reviewer #2:

The authors have addressed well about reviewers' comments. This is revised manuscript can be accepted.

Response: We are pleased to receive your agreement on our response in the last version and thank you very much for your recommendation.

Reviewer #3:

The authors addressed all concerns raised by this reviewer. Thus, this paper is recommended for publication.

Response: We thank the reviewer very much for your recommendation.